# Deep Integration Between Polarimetric Forward-Transmission Fiber-Optic Communication and Distributed Sensing Systems

**DOI:** 10.3390/s24216778

**Published:** 2024-10-22

**Authors:** George Y. Chen, Ming Chen, Xing Rao, Shangwei Dai, Runlong Zhu, Guoqiang Liu, Junhong Lu, Hanjie Liu, Yiping Wang

**Affiliations:** 1Shenzhen Key Laboratory of Photonic Devices and Sensing Systems for Internet of Things, Guangdong and Hong Kong Joint Research Centre for Optical Fibre Sensors, State Key Laboratory of Radio Frequency Heterogeneous Integration, Shenzhen University, Shenzhen 518060, China; gychen@szu.edu.cn (G.Y.C.); 2210452010@email.szu.edu.cn (M.C.); 2250453012@email.szu.edu.cn (X.R.); 2350453003@email.szu.edu.cn (S.D.); 2300453015@email.szu.edu.cn (R.Z.); 2400233054@mails.szu.edu.cn (G.L.); 2410232076@mails.szu.edu.cn (J.L.); liuxiaoniu@whut.edu.cn (H.L.); 2Shenzhen Key Laboratory of Ultrafast Laser Micro/Nano Manufacturing, Key Laboratory of Optoelectronic Devices and Systems of Ministry of Education/Guangdong Province, College of Physics and Optoelectronic Engineering, Shenzhen University, Shenzhen 518060, China; 3Guangdong Laboratory of Artificial Intelligence and Digital Economy (SZ), Shenzhen 518107, China

**Keywords:** distributed sensing, vibration sensing, fiber-optic sensor, forward transmission, PolSK, communication–sensing integration, long-distance

## Abstract

The structural health of fiber-optic communication networks has become increasingly important due to their widespread deployment and reliance in interconnected cities. We demonstrate a smart upgrade of a communication system employing a dual-polarization-state polarization shift keying (2-PolSK) modulation format to enable distributed vibration monitoring. Sensing can be conducted without hardware changes or occupying additional communication bandwidth. Experimental results demonstrate that forward transmission-based distributed vibration sensing can coexist with PolSK data transmission without significant deterioration in performance. This proof-of-concept study achieved a sensitivity of 0.4141 μV/με with a limit of detection (LoD) of 563 pε/Hz^1/2^@100 Hz. The single-span sensing distance can reach up to 121 km (no optical amplification) with a positioning accuracy as small as 874 m. The transmission rate is 300 Mb/s, the *Q_dB_* is 16.78 dB, and the corresponding BER is 5.202 × 10^−12^. For demonstration purposes, the tested vibration frequency range is between 100 and 200 Hz.

## 1. Introduction

With the rapid development of high-speed data applications such as 5G, cloud computing, the Internet of Things, and smart cities, fiber-optic communication networks are heavily utilized. The area covered by optical fiber deployment and the density of networks have surged in the past decade, extending across rural areas, towns, and metropolitan regions [1,2]. Polarization shift keying (PolSK) is a well-known digital modulation technique in the field of communications. Compared to other modulation formats, PolSK has two advantages: (1) It uses the state of polarization (SOP) as a carrier for transmitting information. This greatly reduces nonlinear effects caused by fluctuations in optical intensity within the optical fiber. (2) It supports higher-order modulation and multiple polarization state encoding, which can increase the capacity of transmission [3]. The structural health of communication optical cables and the condition assessment of civil engineering structures hosting these cables are of critical concern for long-term operation and strategic maintenance planning.

Distributed fiber-optic sensing (DFOS) technologies utilize the optical fiber as both the signal transmission medium and the sensing element [4]. By detecting variations in the intensity, phase, and polarization state of the propagating optical signals, DFOS enables continuous distributed measurement of perturbations along the fiber [5,6,7]. Consequently, it has become an ideal non-destructive structural health monitoring method for large-scale infrastructures in various fields such as energy, power, aerospace, telecommunications, transportation, and security.

The reported schemes for fiber-optic communication and sensing integration can be categorized by backscattered light [8,9,10] and forward-transmitted light [11,12,13,14]. The former approach results in different optical propagation characteristics, which requires the separate treatment of the communication and sensing signals. This often leads to the deployment of new optical cables and supporting hardware, raising the cost. The latter approach features very similar optical propagation characteristics, which means that optical fiber and hardware (laser source, amplifiers, and demodulator) can be shared, giving rise to lower-cost deep integration. The deep integration of forward transmission distributed sensing and communication facilitates the sharing of the same laser source, optical fiber, wavelength channel, optical signal, and demodulator. Furthermore, sensing is non-intrusive since it only analyzes the vibration information that already exists within the received communication signal, without changing any hardware or taking up additional communication bandwidth. In 2021, researchers from the California Institute of Technology proposed a polarization-demodulation-based sensing system based on Google’s Curie submarine optical cable. This system realized earthquake and tidal monitoring over a 10,500 km multi-span optical cable by monitoring the state of polarization (SOP) of optical communication traffic [15]. However, this demonstration did not support vibration positioning or distributed vibration mapping. In the same year, collaborating researchers from the University of L’Aquila in Italy presented an analytical interpretation of the mechanisms behind SOP monitoring [16]. Yan et al. also reported long-distance distributed vibration sensors based on forward transmission [17]. Unlike the polarization demodulation used in this work, their work used the FS-ODL method based on frequency shift and coherent detection.

To address the lack of positioning ability for polarization-based forward-transmission distributed fiber-optic vibration sensors, which can be used to realize PolSK communication–sensing integration, we demonstrate a new sensing scheme based on polarization demodulation, forward transmission, and phase-spectrum time delay. To the best of our knowledge, this is the first reported study of deep integration between polarization-based fiber-optic communication and forward-transmission distributed fiber-optic sensing. The shared modules include the laser source, optical fiber, wavelength, optical signal, and demodulator. Additionally, the longest single-span sensing distance (~121 km) is demonstrated for fiber-optic communication–sensing integration. The sensing distance is mainly limited by nonlinear effects such as stimulated Brillouin scattering, as well as system and environmental noise, which both lower the signal-to-noise ratio (SNR) and thus the maximum sensing distance.

## 2. Demodulation Principles

### 2.1. Modulation Format

PolSK can generally be categorized into dual-polarization-state polarization shift keying (2-PolSK) and multi-polarization-state polarization shift keying (M-PolSK) [18]. The fundamental form is the 2-PolSK format, utilizing the SOPs at the two ends of the Bloch sphere to represent binary 0 and 1, since they are the easiest to distinguish. After transmission through an optical fiber, the SOP of the received optical signal is analyzed using a polarization beam splitter and demodulation algorithm. M-PolSK uses two or more SOPs, allowing multiple bits of information to be transmitted simultaneously.

The transmitter module of a typical 2-PolSK system is illustrated in Figure 1a. The laser source outputs a fixed SOP, which is then decomposed into two orthogonally polarized beams via a polarizing beam splitter (PBS). These two optical paths are modulated in parallel through an amplitude modulator (AM), which uses two intensity levels to modulate the input light, in order to represent the binary states “0” and “1”. After modulation, a polarization beam combiner (PBC) merges the two optical signals into a single optical signal with orthogonal polarizations, which now carries the encoded information. This approach enhances the capacity of the communication system. 

The integrated communication–sensing system is shown in Figure 1b, containing both the transmitter and receiver modules. This integration enables distributed sensing to be performed using communication hardware, without adding, removing, or changing any part. The communication system transmits and receives data, which contains vibration information encoded into the optical signal via the optical fiber. The signal demodulation process involves first obtaining the detected parameter (e.g., differential voltage); then using the fast Fourier transform and bandpass filters to separate the low-frequency vibration signals and the medium–high-frequency communication signals; and then performing decision making using predetermined thresholds or calibration curves.

The receiver module of the 2-PolSK system is illustrated in Figure 1c. The combined signal passes through a PBS, where it is again decomposed into two orthogonally polarized components. Each optical path is received by two photodetectors (PDs). These detectors convert the optical signals into electrical signals, with the voltage being proportional to the optical intensity, followed by a subtractor circuit. Alternatively, a balanced photodetector (BPD) can be used, and the output differential voltage signal is proportional to the change in the polarization azimuth. Frequency-domain filtering can separate the sensing and communication signals. Through threshold detection, binary values are assigned to reconstruct the original communication data.

The deep integration method in this work should be able to extend to stacked modulation methods [19]. There should be no significant overlap even when the modulation type is the same between communication and sensing, since the sensing aspect occupies low frequencies while the communication aspect uses much higher frequencies (albeit with some lower frequency components due to the abrupt edges of the waveform).

In terms of multiplexing technology, the deep integration method can still be employed. For example, wavelength-division multiplexing (WDM) and space-division multiplexing (SDM) can be deployed in the same system [20] with sensing. The sensing aspect only needs to use one channel of a multiplexing scheme, such as a single wavelength of a single optical path, and sensing can co-exist with communication without occupying additional bandwidth due to the nature of deep integration that uses a digital copy for separate data analysis.

### 2.2. Demodulation Mechanism

Assuming that the electric field of the received optical signal can be expressed as a superposition of two orthogonal polarization components, *E_x_* and *E_y_*:(1)$$E_{x}=E_{0}cos\theta ,$$
(2)$$E_{y}=E_{0}sin\theta ,$$
where *E*_0_ is the total electric field amplitude of the incident light, and *θ* is the polarization angle of the incident light.

The voltage signals outputs are proportional to the received optical intensities.
(3)$$\;V_{x}=kI_{x}=kE_{x}^{2},$$
(4)$$V_{y}=kI_{y}=kE_{y}^{2},$$
where *I_x_* and *I_y_* are the optical intensities along the *x* and *y* directions, and *k* is a proportionality constant.

The output voltage *V_out_* of the BPD is the difference between the output voltages of the two detectors:(5)$$V_{out}=V_{x}-V_{y}=k\left(E_{0}^{2}{cos}^{2}\theta -E_{0}^{2}{sin}^{2}\theta \right),$$

For small angles relative to the *x*-axis, further simplification yields:(6)$$\;V_{out}\approx kE_{0}^{2}cos\left(2\theta \right)=V_{max}cos(2\theta ),$$

Hence, the output differential voltage signal from the BPD is proportional to the maximum voltage (*V_max_*) observed and the cosine function of the polarization azimuth.

## 3. Experiment Setup

The integrated system combining PolSK and forward-transmission distributed vibration sensing is shown in Figure 2. A linearly polarized coherent laser (Thorlabs SFL1550S, Newton, NJ, USA) with a center wavelength of 1550 nm, a linewidth of 50 kHz, and an output power of 17 mW was employed to probe the sensing system. A polarization controller (PC) (Thorlabs FPC560, Newton, NJ, USA) was employed to ensure a 45° entry into a PBS (Thorlabs PFC1550A, Newton, NJ, USA) to evenly split the optical power. EOM1 (Conquer KG-AMBox, Shen Zhen, China) and EOM2 (AFR F10-0, Zhu Hai, China) independently modulate the intensity of the two optical paths, which is then recombined into a single optical path with orthogonal polarizations by a PBC (Thorlabs PFC1550A, Newton, NJ, USA). Subsequently, the light is launched into a non-circulating fiber loop to form a single-ended configuration (a double-ended configuration can use a straight fiber). One beam propagates clockwise through the first circulator (Thorlabs 6015-3-APC, Newton, NJ, USA), passes through the sensing fiber, and exits through the second circulator. Similarly, the other beam propagates counterclockwise. A set of PBS is used at the two detection ends to separate the polarization axes and convert into differential voltage signals by a pair of BPDs (Thorlabs PDB470C, Newton, NJ, USA). The bandwidth of the BPD is 400 MHz. The signals are digitized by a USB oscilloscope (PicoScope 6428E-D, St. Neots, UK) connected to a computer. Digital bandpass filters are used to isolate the sensing and communication signals for separate analysis. A piezoelectric transducer (PZT) (Coremorrow H01.80, Harbin, China) with 50 m length of coiled fiber is used to create vibrations along certain sections of the sensing fiber. Due to the different optical path lengths between each PZT and the two detectors, the difference in the time-of-flight of a particular frequency can be used to determine the vibration position. This experiment was conducted in an environment where temperature, pressure, and other environmental factors were stable to mitigate any cross-sensitivity with external effects such as ambient temperature. In the future, multiple optical channels with different sensitivity coefficients will be used to simultaneously resolve vibration and temperature changes.

Assuming the fiber length is *L*, when vibration is applied to the sensing fiber, the time delay Δ*t* between the arrival of the signals can be expressed as a function of the vibration position *x*:(7)$$\;∆t=\frac{L-x}{\frac{c}{n}}-\frac{x}{\frac{c}{n}}=\frac{n}{c}\left(L-2x\right),$$

The vibration position *x* can be obtained after rearranging Equation (7):(8)$$\;x=\frac{L}{2}-\frac{c∆t}{2n},$$
where *c* is the propagation speed of light in vacuum, and *n* is the effective index of the fiber core.

## 4. Experiment Results and Discussion

### 4.1. Distributed Vibration Sensing

The relationship between the differential voltage signal and strain was characterized through tests with different PZT strains, and a linear trend was observed, as shown in Figure 3.

The sensitivity taken from the slope gradient of Figure 3 is 0.4141 μV/με. Due to the standard deviation of the intensity noise being 1.4124 μV, the limit of detection (LoD) [21] can be calculated as follows:(9)$$LoD=3.3*\frac{std\left(noise\right)}{sensitivity}=\frac{3.3*1.4124\;}{0.4141}\;\mu \varepsilon =11.26\;\mu \varepsilon ,$$

The normalized LoD, obtained by dividing the LoD by the square root of the detection bandwidth (400 MHz), is 563 pε/Hz^1/2^.

To verify vibration positioning accuracy, a commercial optical time-domain reflectometer (OTDR) was connected to a nominal end of the single-mode fiber (SMF1), and the center position of the optical fiber coiled around the PZT was measured to be at a distance of 100.51 km from the launch end. The total length of the fiber is 120.78 km. Figure 4a shows the differential voltage signals from the two detection ends, while Figure 4b presents the cross-correlation [22] results between the two detection ends. Figure 4c provides an enlarged view around the peak value. The maximum peak occurs at −406.94 µs, and the vibration position is determined to be 101.04 km using Equation (9). Figure 4d displays the positioning results.

When multiple vibration events occur along the sensing fiber, the traditional cross-correlation method treats the vibrations as a collective whole, and thus suffers from significant inaccuracies. Instead, a phase-spectrum time delay method based on frequency-domain analysis is used for resolving each vibration source of different frequencies [23]. Two PZTs were placed at distances of 50.32 km and 100.56 km, with frequencies of 160 Hz and 100 Hz, respectively. The resulting differential voltage signals are shown in Figure 5a. The time delays of 0.4205 ms and −0.0928 ms shown in Figure 5b are calculated using the initial phase of each frequency component of the fast Fourier transform. They correspond to vibration positions of 51.11 km and 102.39 km. In the contour map shown in Figure 5c, the vibrations are resolved as a function of distance, frequency, and amplitude (intensity).

To investigate the accuracy of vibration positioning within the context of integrated sensing applications, a specific vibration point was selected at a distance of 100.51 km, and 100 consecutive measurements were carried out. The piezoelectric transducer (PZT) was driven by a sinusoidal signal with an amplitude of 24 V and a frequency of 100 Hz. The measurement results are shown in Figure 6, which presents the histogram of the vibration position and its probability distribution.

To quantify the positioning accuracy, the root mean square error (RMSE) and standard deviation (STD) were used as metrics. The RMSE can reveal the error between observed values and true values, while the STD is better suited to show the dispersion of a dataset, given by the following expressions:(10)$$X_{RMSE}=\sqrt{\sum_{i=1}^{k}\frac{{\left(x_{m,i}-x_{true}\right)}^{2}}{k}},$$
(11)$$X_{STD}=\sqrt{\sum_{i=1}^{k}\frac{{\left(x_{m,i}-x_{mean}\right)}^{2}}{k}},$$
where *k* is the number of measurements, i.e., *k* = 100, *x_true_* is the true position of the vibration, and *x_mean_* is the mean of the measured position *x_m_*. The RMSE calculated from the above expression is 1.15 km, while the STD is 0.874 km. The peak of the Gaussian fitting curve is located at 100.837 km, which is in close agreement with the actual vibration position (100.51 km) obtained via OTDR, yielding an alternative positioning error of 327 m. The fiber length was verified using a commercial optical time-domain reflectometer (Anritsu MT9085, Atsugi, Japan) and cross-checked with the manufacturer’s datasheet. Note that the absolute difference is not used as the positioning accuracy since it can be calibrated based on the effective index of the fiber.

### 4.2. Communication Operation

The analysis of a PolSK communication test signal is conducted, where the pair of EOMs generates two orthogonal polarization states to represent the data “1” and “0”. In the experiment, EOM1 transmits “01101” and EOM2 transmits “10010”. After modulation, the information is transmitted through the optical fiber, and the communication signal received by the BPD of each detection end is shown in Figure 7. To decide between high and low levels, differential voltages greater than 40 mV are judged as “1”, and those less than 40 mV are judged as “0”. It is seen from Figure 7a and Figure 7b that the periodic communication signals “01101” and “10010” can be clearly distinguished, corresponding to the modulation sequence “01101” from EOM1 and “10010” from EOM2, respectively. Due to the high extinction ratio of the modulation signal (static extinction ratio > 12 dB), even under the influence of vibrations, the communication signal can still maintain a relatively low bit error rate (BER).

The BER can be estimated by analyzing the Q-parameter:(12)$$BER=\frac{1}{2}erfc\left(\frac{Q}{\sqrt{2}}\right),$$
where the complementary error function (erfc) is defined as:(13)$$\;erfc\left(x\right)=\frac{2}{\sqrt{\pi }}\int_{x}^{\infty }e^{-t^{2}}dt,$$

The Q-parameter is defined as the ratio of the signal dynamic range to noise floor. A higher *Q*-parameter indicates a lower BER, given by:(14)$$Q=\frac{V_{top}-V_{bottom}}{\sigma_{1}+\sigma_{2}},$$
where *V_top_* and *σ*_1_ represent the signal amplitude and noise standard deviation for the high level, respectively, and *V_bottom_* and *σ_2_* represent the signal amplitude and noise standard deviation for the low level, respectively.

The *Q*-parameter can also be expressed in decibels (dB), commonly denoted as *Q_dB_*, and is calculated by converting the linear scale into logarithmic scale. The conversion is as follows:(15)$$Q_{dB}=20\log_{10}Q,$$

To analyze the stability and channel quality of the PolSK data transmission under vibrations, an eye diagram is plotted in Figure 8. In the eye diagram, the “1” level (*V_top_*) and “0” level (*V_bottom_*) represent the binary value of 1 or 0, respectively. The average value of the high level (*V_top_*) is 63.74 mV, with a standard deviation (*σ*_1_) of 3.92 mV, while the average value of the low level (*V_bottom_*) is 2.65 mV, with a standard deviation (*σ*_2_) of 10.64 mV. According to Equations (12)–(15), the calculated *Q_dB_* is 12.46 dB, and the estimated BER is 1.359 × 10^−5^.

### 4.3. Integration Analysis

The mutual impact between communication and sensing signals is studied by observing and comparing the system performance under different operational states. Figure 9 reveals the outcome from four types of experiments, to systematically compare the system performance when either the EOM (communication) or PZT (vibration sensing) is active. Figure 9a shows the system noise in the absence of both high-frequency intensity modulation (PolSK) and low-frequency intensity modulation (vibration). The noise is mainly attributed to the random changes in the SOP caused by fluctuations in the ambient environment. Figure 9b shows the PolSK modulation without vibration perturbations, which reveal an extinction ratio of 5.79 dB, a *Q_dB_* of 16.78 dB, and a BER of 5.202 × 10^−12^. Figure 9c shows the detection of sinusoidal vibration signals without PolSK modulation, with an LoD of 564 pε/Hz^1/2^ and a positioning accuracy (STD) of 766 m. Finally, Figure 9d features the existence of both PolSK modulation and vibration signals, which gives rise to an extinction ratio of 6.02 dB (trough of vibration envelope) or 7.23 dB (peak of vibration envelope), a *Q_dB_* of 12.46 dB, and a BER of 1.359 × 10^−5^. The sensing performance features an LoD of 621 pε/Hz^1/2^ and a positioning accuracy (STD) of 874 m.

It is clear that communication has a degree of impact on the sensing of vibrations, through comparing LoD (OFF 564 pε/Hz^1/2^ vs. ON 621 pε/Hz^1/2^) and STD (OFF 766 m vs. ON 874 m) between Figure 9c and Figure 9d. As for the impact of vibrations on communications, by comparing the extinction ratio (OFF 5.79 dB vs. ON 6.02–7.23 dB), *Q_dB_* (OFF 16.78 dB vs. ON 12.46 dB), and BER (OFF 5.202 × 10^−12^ vs. ON 1.359 × 10^−5^) between Figure 9b and Figure 9d, it is clear that external vibrations deteriorate the data transmission quality. BER deterioration can affect data integrity, throughput and efficiency, error correction overhead, user experience, system design complexity, reliability and availability, and regulatory compliance. To regulate BER within acceptable limits, engineers often employ a variety of techniques, including using stronger error correction coding, higher SNR through better hardware design, adaptive modulation schemes, and robust synchronization between the transmitter and receiver. It needs to be stressed that communication signals usually already have vibration information encoded as noise, and the addition of the distributed sensing ability does not add extra noise. Furthermore, the sensing function is non-intrusive, as it does not change hardware nor occupy additional communication bandwidth.

Although the bit rate (300 Mb/s) demonstrated in this proof-of-concept study is low compared to commercial fiber-optic communication systems, the viability of deep integration between polarization-based communication and sensing was verified. The transmission rate can be massively increased via additional modulation formats with wavelength- and space-division multiplexing.

## 5. Conclusions

A proof-of-concept of deep integration was demonstrated between fiber-optic communication using 2-PolSK and distributed fiber-optic vibration sensing based on forward transmission. No hardware changes are necessary to add distributed sensing functionality to the communication system. Sensing can be conducted without occupying additional communication bandwidth, due to the non-intrusive extraction of vibration information from received data. Data transmission and multi-point vibration sensing were characterized for signal recovery and crosstalk impact. It was observed that the *Q_dB_* parameter of communication decreased (−4.3 dB) in the presence of vibrations, though the sensing function merely analyzes what is already present in the received communication signal, and thus does not affect the quality of transmission. The sensing performance also declined slightly (−14% in positioning accuracy and −10% in LoD) when data transmission was enabled. This is attributed to the low-frequency components of the square-wave modulation overlapping with the low-frequency vibration signals. This work demonstrated a sensitivity of 0.4141 μV/με with a limit of detection (LoD) of 563 pε/Hz^1/2^@100 Hz. The single-span sensing distance can reach up to 121 km (no optical amplification) with a positioning accuracy as small as 874 m. The transmission rate is 300 Mb/s, the *Q_dB_* is 16.78 dB, and the corresponding BER is 5.202 × 10^−12^. For demonstration purposes, the tested vibration frequency range is between 100 Hz and 200 Hz. This work helps pave the way for high-speed fiber-optic communications with built-in distributed monitoring abilities. 

## Figures and Tables

**Figure 1 sensors-24-06778-f001:**
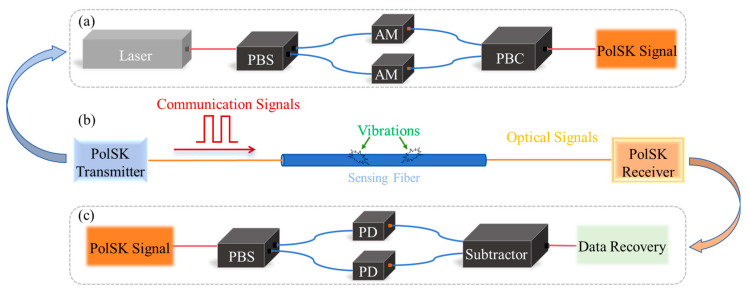
Concept of PolSK communication and sensing deep integration. (**a**) PolSK transmitter module; (**b**) optical transmission with vibration influence; (**c**) PolSK receiver module. AM: amplitude modulator; PBS: polarization beam splitter; PBC: polarization beam combiner; PD: photodetector.

**Figure 2 sensors-24-06778-f002:**
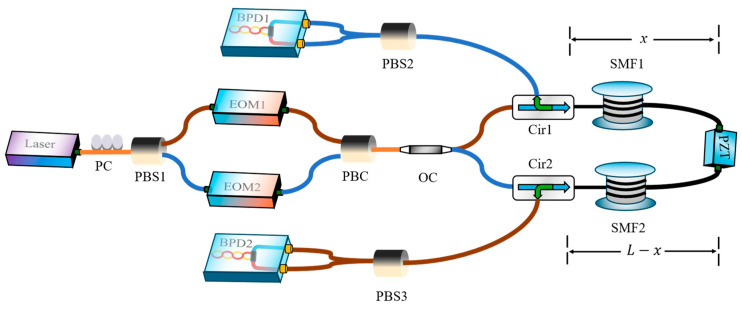
Schematic of the integrated system. PC: polarization controller; EOM: electro-optic modulator; OC: optical coupler; Cir: circulator; SMF: single-mode fiber; BPD: balanced photodetector.

**Figure 3 sensors-24-06778-f003:**
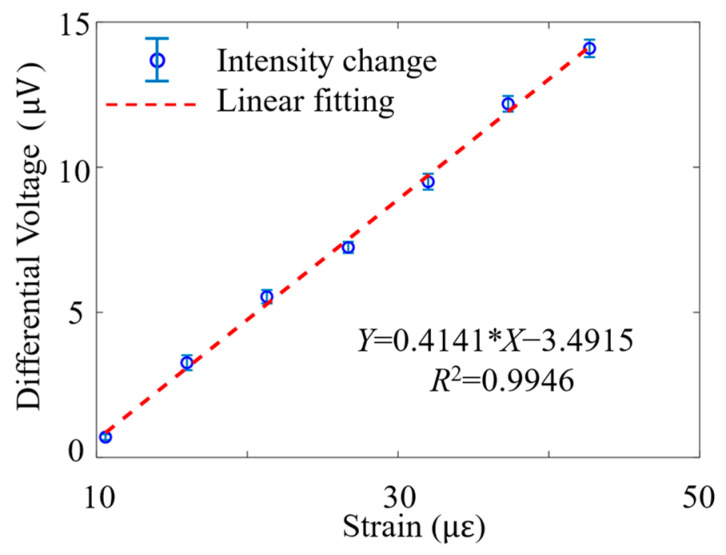
Relationship between applied strain and differential voltage.

**Figure 4 sensors-24-06778-f004:**
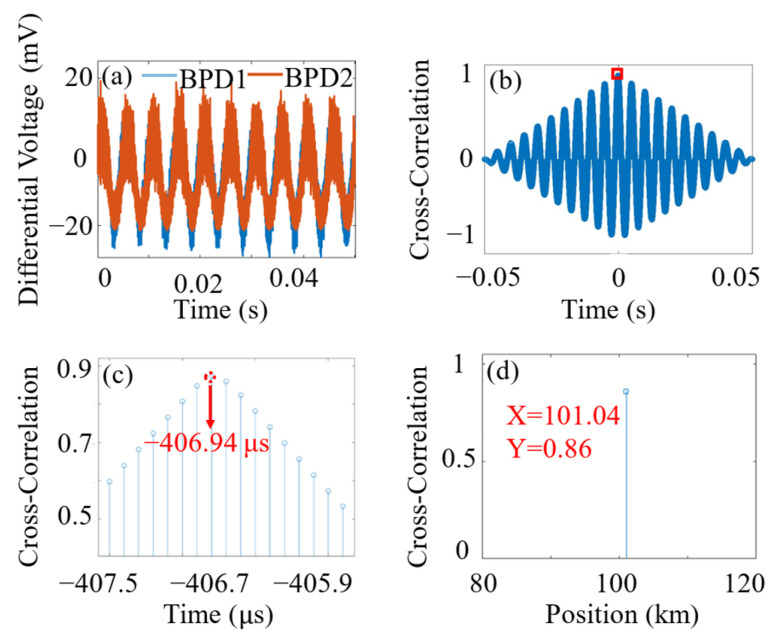
Single-point vibration positioning processing sequence. (**a**) Differential voltage signal; (**b**) cross-correlation results, red square: indicates area for magnification; (**c**) local magnification around the highest peak; (**d**) corresponding vibration position. The PZT was driven with a 200 Hz sinusoidal signal.

**Figure 5 sensors-24-06778-f005:**
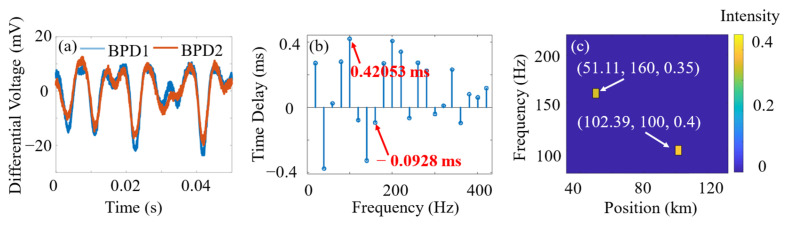
Multi-point vibration positioning processing sequence. (**a**) Differential voltage signal; (**b**) time-delay spectrum showing corresponding positions associated with signal frequencies (identified from amplitude spectrum); (**c**) 3D relationship between vibration position, frequency, and amplitude.

**Figure 6 sensors-24-06778-f006:**
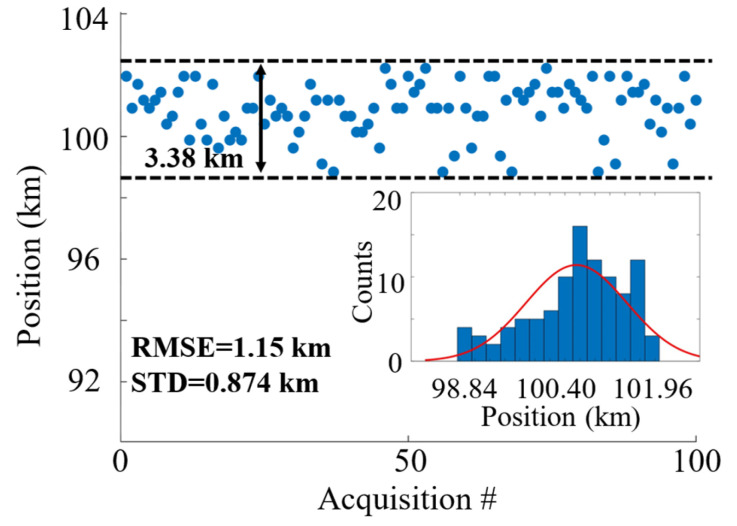
Vibration positioning accuracy from 100 repeated measurements. Inset: histogram of the positioning results with a Gaussian fitting.

**Figure 7 sensors-24-06778-f007:**
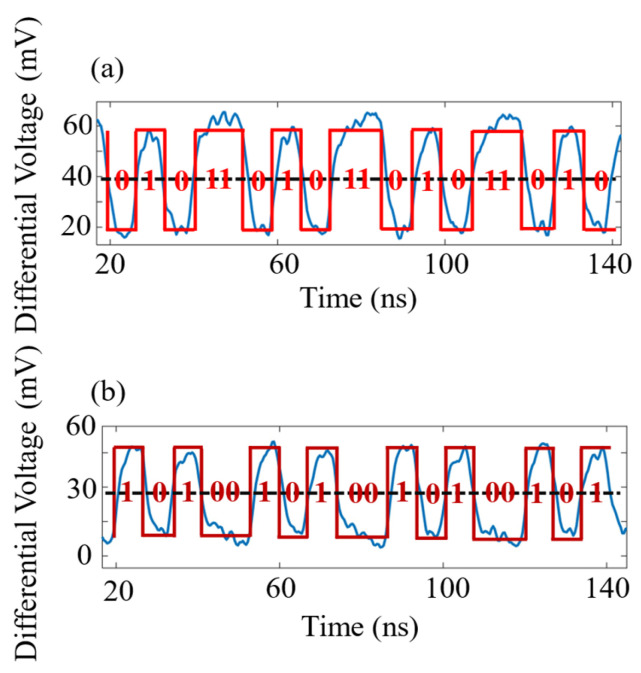
PolSK communication bit recovery under the influence of vibrations. (**a**) BPD1 signal (01011 sequence); (**b**) BPD2 signal (10100 sequence). Red line: recovery of original communication signal; blue line: BPD signal; black dashed line: decision threshold.

**Figure 8 sensors-24-06778-f008:**
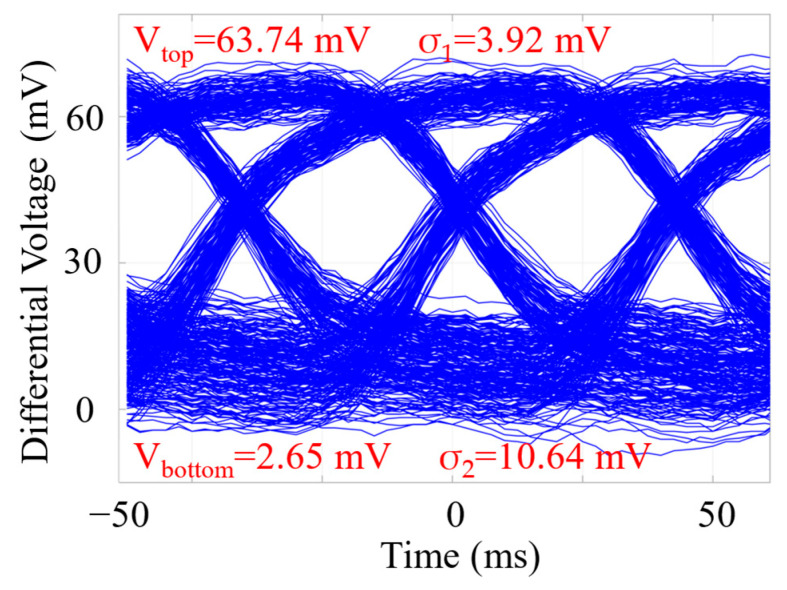
Eye diagram of PolSK communication–sensing system under the influence of vibrations.

**Figure 9 sensors-24-06778-f009:**
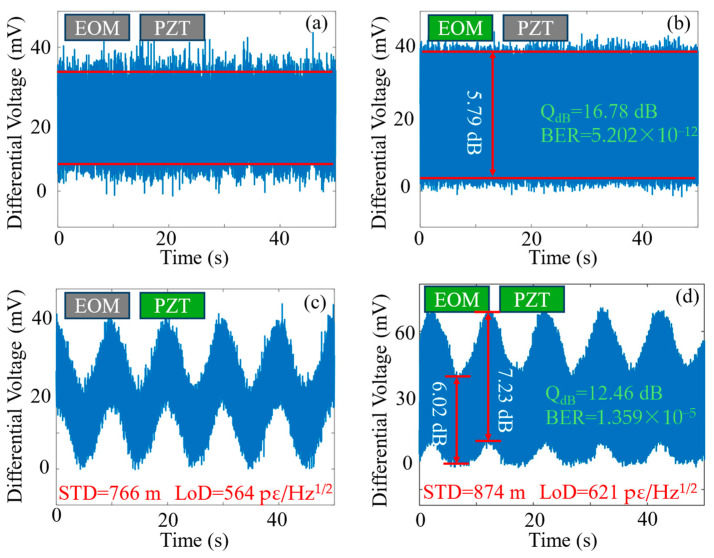
Communication–sensing crosstalk analysis. (**a**) No data transmission and vibration; (**b**) data transmission only; (**c**) vibration only; (**d**) data transmission with vibration. The vibration amplitude applied was 26.7 με. Red line: minimum to maximum value.

## Data Availability

Data underlying the results presented in this paper are not publicly available at this time but may be obtained from the authors upon reasonable request.

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
