# Peer review of "Deep Integration Between Polarimetric Forward-Transmission Fiber-Optic Communication and Distributed Sensing Systems"

_sensors, 2024, doi:10.3390/s24216778_

Round 1
Reviewer 1 Report
Comments and Suggestions for Authors
General comments to the Author.
In this paper, the authors present a paper titled: “Deep Integration between Polarimetric Forward-Transmission Fiber-Optic Communication and Distributed Sensing System”. In the manuscript, the authors propose and demonstrate an interesting sensing scheme based on polarization demodulation, forward transmission, and phase-spectrum time delay. This study shows an integration between polarization-based fiber-optic communication and forward-transmission distributed fiber-optic sensing. The longest single-span sensing distance, around 121 km, is demonstrated for fiber-optic communication-sensing integration.
Overall, I consider the paper to present a notable description of a distributed sensing system, which is according to the scope of the Sensors. However, some minor aspects of the manuscript need to be clarified and adjusted before it is authorized for publication.
To improve the overall quality of the paper, there are a few suggestions and remarks reported in what follows:
1. Figure 1 (page 2) shows the communication diagram based on the PolSK scheme. However, due to the logical sequence of the text, the reader would expect the explanation of the AM component and its function to be detailed in lines 90-95, but this is not the case. It is only mentioned in the figure caption. I suggest that this aspect be corrected.
2. Figure 3, page 5. The authors show the relationship between applied strain and differential voltage. Why is it not possible to show an additional point close to 10 με to provide more data? Furthermore, the authors do not explain the repeatability of this curve. This is essential because it defines the detection limit of the proposed system.
3. In line 180, page 5, the authors affirm: “To demonstrate single-point vibration positioning, a PZT was placed at a distance of 180 100.51 km from a nominal end of the sensing fiber, verified using a commercial optical 181 time-domain reflectometer (OTDR)”. However, it is not clear in which arm of the system the OTDR is connected to verify the single-point vibration positioning of the PZT. This aspect needs to be clarified.
4. On the right side of the system shown in Figure 2, the experimental configuration is like a Sagnac ring interferometer. This array is sensitive to external agents. How to compensate for or attenuate additional effects such as temperature?
Reviewer 2 Report
Comments and Suggestions for Authors
The authors present a form of deep integration between fiber-optic communication (PolSK) and polarization-based forward transmission distributed fiber-optic vibration sensor. This work is useful and cost-effective for upgrading existing communication cables with sensing capabilities. Some issues need be addressed more clearly:
(1) The vibration sensing based on forward transmission has been reported in the literature(DOI: 10.1109/JLT.2020.3044676), and it is recommended to compare the method proposed in this paper with it.
(2) What limits the sensing distance? What is the range of measurable vibration frequencies?
(3) Some figures issues: The points of interest in Fig. 5c are not clear enough. The coordinate line in Fig.7 is missing. There is a lot of information in Fig. 9, I suggest changing some of the text labels to icons or different formatting styles to distinguish them better.
(4) Can the deep integration method be extended to stacked modulation methods and multiplexing techniques? If so, please elaborate.
(5) I noticed that when there is a vibration event, the BER of the system deteriorates significantly. In actual communication, how much impact does such BER deterioration have on communication?
Reviewer 3 Report
Comments and Suggestions for Authors
The originality of the manuscript under review is in amalgamating an all-optical approach for of a communication system employing a dual-polarization-state polarization shift keying (2-PolSK) modulation format as well as distributed sensing system to enable distributed vibration monitoring.
In the manuscript, a new sensing scheme based on polarization demodulation, forward transmission and phase-spectrum time delay is demonstrated and verified by the theory and experiments. To the best of my knowledge, this is the first-reported study of deep integration between polarization-based fiber-optic communication and forward-transmission distributed fiber-optic sensing, where the authors paved the way for high-speed fiber-optic communications with built-in distributed monitoring abilities. The results show good agreement between the theoretical and experimental data in terms of the longest single-span sensing distance and the quality of the high-speed transmitted signals.
The manuscript is well written in terms of the quality of presentation and readability, so I recommend accepting it after minor revision to improve readability.
1. When describing the experiments, it is advisable to indicate not only the optical time-domain reflectometer model (lines 229-230), but also the models of the optoelectronic and optical components included in the circuit (Figure 2), as well as the digital oscilloscope used to measure the eye diagram (Figure 8) and the communication-sensing crosstalk analysis (Figure 9).
2. Subsection 4.2. In the text (lines 240-241): “It is evident from Figure 7a and Figure 7b that the communication signal “01101” and “10010” can be clearly distinguished”. Nevertheless, these Figures show different sequences. I realize that they are only shifted by 2 clock periods, but I think this fact should be somehow noted in the text or in the Figures.
